# Understanding the Molecular Mechanisms Underlying the Pathogenesis of Arthritis Pain Using Animal Models

**DOI:** 10.3390/ijms21020533

**Published:** 2020-01-14

**Authors:** Jeong-Im Hong, In Young Park, Hyun Ah Kim

**Affiliations:** 1Division of Rheumatology, Department of Internal Medicine, Hallym University Sacred Heart Hospital, Gyeonggi 14068, Korea; jhi1126@naver.com (J.-I.H.); hoyo012@hanmail.net (I.Y.P.); 2Institute for Skeletal Aging, Hallym University, Chuncheon 24252, Korea

**Keywords:** osteoarthritis, rheumatoid arthritis, animal model, pain

## Abstract

Arthritis, including osteoarthritis (OA) and rheumatoid arthritis (RA), is the leading cause of years lived with disability (YLD) worldwide. Although pain is the cardinal symptom of arthritis, which is directly related to function and quality of life, the elucidation of the mechanism underlying the pathogenesis of pain in arthritis has lagged behind other areas, such as inflammation control and regulation of autoimmunity. The lack of therapeutics for optimal pain management is partially responsible for the current epidemic of opioid and narcotic abuse. Recent advances in animal experimentation and molecular biology have led to significant progress in our understanding of arthritis pain. Despite the inherent problems in the extrapolation of data gained from animal pain studies to arthritis in human patients, the critical assessment of molecular mediators and translational studies would help to define the relevance of novel therapeutic targets for the treatment of arthritis pain. This review discusses biological and molecular mechanisms underlying the pathogenesis of arthritis pain determined in animal models of OA and RA, along with the methodologies used.

## 1. Introduction

Musculoskeletal diseases, including osteoarthritis (OA) and rheumatoid arthritis (RA), are the leading causes of years lived with disability (YLD) worldwide. Furthermore, YLD due to OA increased by 31.5% from 2006 to 2016, in association with the aging of the population [1]. Pain is the cardinal symptom of both OA and RA, which directly affects the decision to seek medical care. In addition, pain is closely related to function and quality of life, such that knee pain is a better predictor of disability than radiographic changes in OA [2,3]. The development of effective therapeutics for optimal pain management has lagged behind other areas, such as inflammation control and the regulation of autoimmunity, which is partially responsible for the current epidemic of opioid and narcotic abuse. A recent report showed that nearly 10% of all opioids prescribed in Australian general practice are prescribed for OA [4]. Similarly, in a survey of Swedish residents aged ≥35 years, 12% of incident opioid dispensations were attributable to OA and/or its related comorbidities [5]. Although there is inconclusive evidence for the benefits of opioids for arthritides and increasing awareness of the risks, opioid prescription rates for OA in the USA remained stable between 2007 and 2014 [6]. On the other hand, despite recent advances in the treatment of RA utilizing effective immunosuppressive therapies based on a better understanding of its underlying mechanism, remaining pain affected almost one third of early RA patients with a good clinical response [4]. By 2014, 41% of patients with RA in the USA were regular users of opioids [7]. Therefore, to optimize the care of patients with OA and RA, the elucidation of the mechanisms underlying the pathogenesis of pain in these diseases is of great importance. Intuitively, pain from arthritis arises from direct nociceptive mechanisms, such as inflammation and structural joint damage. However, in addition to nociception, arthritis pain involves diverse mechanisms, including the processing of pain in the nervous system, as well as psychological distress [8]. Several well-established animal models of OA and RA are available to study the mechanisms underlying the pathogenesis of joint damage and immune/inflammatory regulation. A range of behavioral and neurophysiological approaches have been used for the delineation of pain in animal models of arthritis. Due to the inherent technological challenges in the quantitative assessment of pain in animal models, however, caution is required when attempting to extrapolate discoveries made in animal models to human patients. This review discusses biological and molecular mechanisms underlying the pathogenesis of arthritis pain obtained in animal models of OA and RA along with the methodologies used.

## 2. Osteoarthritis

### 2.1. Pain in Clinical OA

Radiographic changes in OA are poorly correlated with pain and physical function, and the risk factors for radiographic OA are not the same as those for OA pain [9,10,11]. As pain is strongly correlated with person-level psychological, social, and cultural factors aside from joint damage, studies involving human OA subjects have a high risk of being influenced by confounding effects between individual subjects. Studies using a within-person, knee-matched, case-control design minimize such risks by including patients with knees discordant for the presence of pain or pain severity, and have shown that the severity of radiographic knee OA is indeed strongly associated with both the presence of frequent knee pain and severity of pain in diverse ethnic groups [12,13]. On the other hand, a community study showed that a significant number of subjects with Kellgren and Lawrence (KL) grade 4 OA, which is characterized by a marked loss of articular cartilage and profound subchondral bone changes, did not report pain [14]. As hyaline cartilage, the main focus of interest in both clinical and laboratory research on OA, is an aneural tissue, it is unable to generate nociceptive input and is an unlikely source of pain. In an arthroscopic study that probed the human knee without intra-articular anesthesia, a conscious subject reported remarkable sensitivity to the mechanical loading stimulus in anterior synovial tissues, fat pad, and capsule, but not the cartilage [15]. These findings were corroborated by the results of magnetic resonance imaging (MRI) studies correlating pain with synovitis/effusion in OA subjects [16]. It is notable that the sensation experienced with similar probing of the cruciate ligaments and menisci results in variable degrees of pain depending on the location of the probe, while among OA subjects, meniscus degeneration detected in knee MRI is not correlated with pain [17,18]. This discrepancy may be related to differences between study subjects, such as healthy volunteers vs. OA patients. Bone marrow lesions (BMLs), which in MRI scans appear as regions of hyperintense marrow signals, are associated with knee pain as well as the progression of cartilage damage [19,20]. They often occur before joint degeneration is established, and resolution of knee pain is associated with reduced BML size, suggesting that BMLs may serve as a biomarker for OA structural damage as well as pain [16,21]. In a study of two groups of subjects matched by macroscopic cartilage damage score, the expression of nerve growth factor (NGF) in osteochondral channels and osteoclast densities in subchondral bone were higher in a symptomatic than in an asymptomatic group [22]. These observations suggest that subchondral pathology is associated with symptomatic knee OA, independent of cartilage lesions.

In addition to nociceptive mechanisms arising from structural changes, recent evidence indicates that neurophysiological pathways contribute to OA pain, such that persistent tissue damage and inflammation in the joint induce mechanisms of peripheral and central sensitization [23]. In a study involving 2126 subjects with or at risk of knee OA, mechanical temporal summation (an augmented response to repetitive mechanical stimulation and a measure of central sensitization) and pressure pain thresholds (a measure of sensitivity to pain evoked by mechanical stimulation of nociceptors and a reflection of activity-dependent peripheral sensitization) were associated with pain severity but not radiographic OA [24]. Interestingly, the duration of knee OA was not associated with these markers of pain sensitivity, suggesting that individual predisposition to sensitization rather than induction by peripheral nociceptive input from OA pathology may play a role in pain [24]. A study of patients with hand OA revealed that central sensitization was common, and lower local and widespread pressure pain thresholds and the presence of temporal summation were associated with higher hand pain intensity [25]. These results suggest that the sensitization-associated pain phenotype is common to OA in different locations. In a study of 216 patients with different degrees of knee pain, although all subjects showed high physical impairment, low quality of life, and high pain catastrophizing compared to controls, four distinct knee pain profiles with unique combinations of biochemical markers, pain biomarkers, physical impairments, and psychological factors were apparent [26]. These observations suggest that mechanism-based diagnosis and individualized treatment may be possible for the treatment of knee OA pain. In the early stages of OA, pain occurs intermittently in response to specific activities and movement, which is characteristic of nociceptive pain [27]. As OA progresses, however, pain increases, such that it becomes constant and occurs even at rest [28]. There is debate regarding whether the aggravation of pain is indicative of the development of neuropathy arising from damage to the neurons innervating the joint. In two histological studies that used samples of tissue from patients with end-stage OA, relevant changes in joint innervation for the generation of pain, such as tidemark breaching by vascular channels containing sympathetic and sensory nerves, the presence of free nerve fibers within the subchondral bone marrow, and decreased nerve fibers in synovium with inflammation, were demonstrated [29,30]. Clinical trials examining the efficacy of neuromodulating agents, such as serotonin-norepinephrine reuptake inhibitors (SNRIs) and anticonvulsants, showed inconsistent results in pain reduction, however, and none of these agents reached a high level of recommendation in recent OA treatment guidelines [31,32]. NGF induces sensitization of peripheral nociceptive terminals as well as the development of the nervous system, and plays a key role in acute and chronic pain [33]. Biologic agents that specifically block NGF revealed dramatic pain relief among OA patients, and despite their unique side effect profile, such as alterations in peripheral sensation and development of arthropathies, they are in active clinical development [34,35].

### 2.2. Animal Models for Studying the Pathogenesis of OA Pain

#### 2.2.1. Animal Models

Although a variety of OA animal models have been developed, there is no single model that can replicate all aspects of human OA. The most commonly used animal models employ surgical or chemical induction [36]. Surgically induced OA models reflect posttraumatic human OA and include destabilization of the medial meniscus (DMM), anterior cruciate ligament transection (ACLT), and medial meniscal transection (MNX) models. These models are characterized by a rapid onset of the disease, consistency in the development of pathology, and less dependence on genetic background [37,38,39]. The monosodium iodoacetate (MIA)-induced arthritis model, which involves cartilage degeneration by suppression of chondrocyte metabolism, is the most widely used chemical model [40]. Although easy and reproducible, a transcriptome study reported that there were large discrepancies in the transcriptome profile between MIA-induced arthritis and human OA [41]. In addition, initiating events, such as the inflammation and inhibition of the glycolytic pathway, and pathological changes, such as subchondral bone necrosis and collapse, in MIA are not typical of human OA [36,42,43]. To produce animal OA models that represent human pathology more closely, noninvasive models have been developed, including bipedal walking and mechanical joint loading models. The bipedal walking model is induced by obligatory bipedal exercise, and pathogenic changes in this model are consistent with the pathology observed in humans [22]. The mechanical joint loading model is induced through vertically compressive loading on the knee and ankle joints, which shows a pathogenesis similar to human OA [44]. The STR/ort mouse develops spontaneous OA at a young age [45], whereas other mouse strains are relatively resistant to the spontaneous development of OA [46]. The selection of a suitable animal model is required to investigate the mechanism underlying the progression of the disease and pain and to evaluate the efficacy of therapeutics [47].

#### 2.2.2. Behavioral Tests to Assess Pain in OA Animal Models

There are many approaches for assessing the behaviors associated with OA pain in animal models [48]. The most common behavioral assays are the von Frey, weight bearing, hot plate, rotarod, and spontaneous pain-related behavior tests. The von Frey test is used to evaluate mechanical allodynia in mice and rats [49]. Briefly, the rodents are placed individually on a mesh grid and calibrated von Frey filaments are applied to the hind paw in ascending order of size, until the fiber bends by force. The animal response, including withdrawal and shaking or licking the paw, is considered a positive pain response [50]. Static weight bearing is assessed using an incapacitance meter, and the average weight bearing applied to the hind paw between the injured limb and contralateral limb is measured [50]. A recent study showed that the positioning of the animal in the incapacitance test may affect its weight distribution, so care should be taken to minimize the influence of positioning [51]. Catwalk automated gait analyses measure changes in spontaneous gait due to joint pain. Animals pass an illuminated glass platform while gait is recorded with a video camera connected to a computer, and the changes in gait pattern, such as swing speed and stride length, are analyzed [52]. Thermal hyperalgesia is assessed using the hot plate test, where the rodent is placed on a metal surface maintained at a constant temperature. The response time (latency) is measured until licking or shaking of the ipsilateral hind paw and jumping on a hot plate occur [53]. Testing latency should be performed without incurring tissue damage and each trial should be separated by an interval of at least 10 min [54]. The rotarod test is used to evaluate changes in motor skill. The rodent is placed on a rotarod machine with automatic timers and falling sensors, and the rod speed is accelerated gradually from 0 to 40 rpm/min [55]. The moving time until falling off is measured. The number of trials in hot plate and rotarod tests should be minimized to exclude learning effects [48,55]. Finally, the Laboratory Animal Behavior Observation Registration and Analysis System (LABORAS) test measures the spontaneous pain-related behavior of rodents, including ambulation, and exploratory behavior. Animals are housed in the LABORAS cage and their behaviors are recorded for 2–12 h [56,57]. For all pain measurement tests, the influence of stress, such as laboratory environment and testing duration, on the pain response should be taken into consideration.

It is notable that, until recently, the most commonly used induction methods to study OA pain were distinct from those most often used to investigate the pathophysiology and regulation of structural joint damage [36]. DMM and MIA models are the most commonly used models in pain research and show mechanical, thermal, and spontaneous pain behavior occurring at the onset of OA [58,59]. In general, chemically induced models exhibit pain responses evoked with von Frey, incapacitance, and hot plate tests more quickly than surgically induced models. Therefore, pain responses in the MIA model appear rapidly within 1 week post-injection [50,60,61,62,63,64], while more insidious responses are observed in the DMM model. In one study, STR/ort mice, which develop spontaneous OA, did not show any signs of pain, such as mechanical allodynia, cold sensitivity, or joint compression-related vocalization, with the development of OA [65]. This is reminiscent of human OA, in which pain does not develop, despite advanced joint damage. On the other hand, mechanical joint loading-induced OA animals develop a robust pain phenotype, except for thermal hyperalgesia [44]. Table 1 briefly summarizes the pathology and pain behaviors reported in OA models [38,65,66,67,68,69,70,71,72,73,74,75,76,77,78,79,80,81,82,83,84,85,86,87,88,89,90].

#### 2.2.3. Understanding the Molecular Mechanism Underlying the Pathogenesis of OA Pain via Experiments in Animal Models

In animal models, as in humans, the extent of joint damage and pain are not always correlated [36]. For example, in one study that used a rat model of partial MNX- and MIA-induced OA, the majority of pain responses were apparent within 1 week of surgery or iodoacetate injection, whereas gross joint damage was not evident until around day 21, mirroring the clinical situation where the extent of joint damage is not always correlated with pain [91]. In a recent study that used CRISPR/Cas9 technology to ablate OA-associated genes, although ablation of matrix metalloproteinase 13 (MMP-13) or interleukin 1beta (IL-1β) markedly attenuated structural deterioration, the extent of pain relief was modest compared to the ablation of NGF, which significantly increased joint damage, despite its profound analgesic effect [92]. In MNX- and MIA-induced OA of rats, it was shown that pain responses to NGF locally administered into the knee cavity were increased in OA compared to non-OA joints, indicating that local production of NGF may be crucial for the generation of pain [62,93]. In a DMM model, the loss of protein kinase C delta (PKCδ) expression prevented cartilage degeneration but exacerbated OA-associated hyperalgesia [94]. Increases in sensory neuron distribution in knee OA synovium and activation of the NGF-tropomyosin receptor kinase (TrkA) axis in innervating dorsal root ganglia were highly correlated with knee OA hyperalgesia, indicating the role of NGF/TrkA signaling in OA pain, independent of cartilage preservation [94]. On the other hand, cartilage degradation products generated during joint damage may directly induce pain. A disintegrin and metalloproteinase with thrombospondin motifs 5 (ADAMTS-5), a critical mediator of cartilage degeneration during the development of OA, was found to mediate pain, because both ADAMTS-5 knockout (KO) mice and anti-ADAMTS-5 antibody in wild-type (WT) mice showed inhibition of pain as well as joint degeneration induced with DMM [95,96]. Hyalectan fragments generated by ADAMTS-5 have been suggested to directly stimulate nociceptive neurons as well as glial activation, promoting increased pain perception [97,98]. Intra-articular injection of the 32-amino-acid (32-mer) aggrecan fragment provoked knee hyperalgesia in a Toll-like receptor 2 (TLR-2)-dependent way, and induced the expression of the proalgesic chemokine (c-c motif) ligand 2 (CCL-2) in dorsal root ganglion (DRG) nociceptive neurons [99]. On the other hand, the release of the proalgesic chemokine monocyte chemoattractant protein 1 (MCP-1) in DRG cultures by S100 calcium-binding protein A8 (S100A8) and alpha-2 -macroglobulin (α2M) was mediated by TLR-4, while TLR-4 KO mice were not protected from mechanical allodynia or from joint damage induced with DMM [100]. Further research is needed to elucidate the biological effects of damage-associated molecular patterns (DAMPs), which act as TLR ligands on pain-sensing neurons, as well as the complex balance between pro- and anti-inflammatory signaling pathways activated by OA DAMPs [101].

OA animal models differ in terms of the inflammatory response evoked in the synovium. In general, chemical-induced models tend to exhibit more inflammation in the synovium compared to surgical models. As synovitis is observed before the thinning of the articular cartilage or subchondral bone changes in the MIA model, the abovementioned augmentation of pain behavior in the MIA model may reflect the influence of inflammation on pain [102]. The collagenase-induced OA (CIOA) model, which is induced by chemical instability resulting from intra-articular injection of collagenase, also exhibits pronounced synovitis and early exhibition of thermal hyperalgesia, manifesting as early as 1 week after injection [61], while DMM mice do not exhibit thermal hyperalgesia until the late phase of the disease [95]. Persistent inflammation with fibrosis in the synovial tissue of MIA model was accompanied by pain avoidance behavior before articular cartilage degeneration and by an increase in calcitonin gene-related peptide (CGRP)-positive fibers in the DRG and synovium [103]. CIOA induced an increase in interferon regulatory factor 4 (IRF-4), CCL-17 and chemokine (c-c motif) receptor 4 (CCR-4), the CCL-17 receptor; gene-deficient mice were protected from pain as well as joint destruction, indicating that these molecules are required for the development of OA pain [104]. The therapeutic neutralization of CCL-17 ameliorated both pain and joint damage, whereas the cyclooxygenase 2 (COX-2) inhibitor only ameliorated pain [104].

Subchondral changes in bone are common in most OA animal models. A recent study showed that osteoclast-initiated subchondral bone remodeling plays an important role in the generation of pain in ACLT-induced animal OA [84]. Netrin-1 secreted by osteoclasts led to sensory nerve axonal growth in subchondral bone, and reduced osteoclast formation, induced by the knockout of receptor activator of nuclear factor kappa-B ligand (RANKL) in osteocytes, inhibited pain behavior in OA mice, suggesting that the modulation of subchondral bone remodeling may have therapeutic potential for OA pain [77]. This result is consistent with findings of clinical trials of drugs that modify subchondral bone. Strontium ranelate, an osteoporosis drug that simultaneously enhances osteoprotegerin (OPG) expression and downregulates RANKL expression in primary human osteoblastic cells, leading to decreased osteoclastogenesis and bone resorption, has been shown to ameliorate pain as well as cartilage volume loss and bone marrow lesions in OA patients [105,106,107]. Bisphosphonates, which inhibit osteoclast activity, are effective for reducing pain and BML size in OA patients [108]. These results demonstrate the potential of finding novel therapeutic molecules for pain by reciprocal use of data derived from both clinical and animal studies.

Obesity has been identified as a key risk factor for the development and progression of OA in numerous epidemiological studies in different ethnic populations. Three studies evaluated the relationship between obesity and OA development via the inhibition of transient receptor potential cation channel, subfamily V, member 4 (TRPV-4), lecithin-cholesterol acyltransferase (Lcat), apolipoprotein A1 (Apoa1), and low-density lipoprotein receptor (Ldlr) in mice fed with a high-fat diet (60% kcal), Western-type diet (4.5 Kcal/g) or cholesterol-rich diet [109,110,111]. OA was spontaneous or collagenase-induced and, overall, these genes were OA-preventing. On the other hand, how obesity affects pain has not been studied in sufficient detail. It is notable that obese leptin-deficient (ob/ob) and leptin receptor-deficient (db/db) mice with leptin signaling impairment do not show increased incidence of knee OA [112]. However, diet-induced obesity significantly increases the severity of OA following intra-articular fracture, with increases in synovitis and inflammatory cytokine IL-12p70 level [113]. On the other hand, in one study, pain behavior measured with thermal hyperalgesia and spontaneous activity tests were not significantly correlated with either weight gain or OA induced by DMM in obese mice fed a high-fat diet [114]. Adenoviral overexpression of the genes encoding the cholesterol hydroxylases cholesterol 25-hydroxylase (CH25H) and 25-hydroxycholesterol 7α-hydroxylase (CYP7B1) in mouse joint tissues caused OA, whereas knockout or knockdown of these hydroxylases abrogated it. In addition, pain induced with DMM in CH25H KO mice was attenuated, suggesting a role of cholesterol metabolism in OA pain [67].

Age is another strong risk factor for OA in human patients. Although it varies by strain and sex, some inbred strains of mice, such as STR/ort and C57BL/6, spontaneously develop OA with age, and spontaneous joint damage does develop in the majority of mouse strains >12 months of age [115]. Although pain is common among elderly people due to a variety of causes, such as age-related musculoskeletal diseases, comorbid illness, and psychological changes, unlike the structural changes in OA, aging has not been shown to be associated with pain in OA subjects. This is in contrast to reports showing age-related peripheral and central nociceptive changes, such as a decreased density of unmyelinated fibers [116], widespread degenerative changes in spinal dorsal horn sensory neurons [117], and declines in the neural opioid and non-opioid analgesic mechanisms mediating endogenous pain inhibitory systems [118]. Systemic inflammation was shown to correlate with pain in older adults with knee OA, such that an increase in serum tumor necrosis factor alpha (TNF-α) and high sensitivity C-reactive protein (hs-CRP) corresponded to increased knee pain and high levels of the soluble receptors for TNF-α correlated with decreased physical function [119,120,121]. On the other hand, mice with a deletion of the IL-6 gene were found to have more severe age-related OA compared to age-matched WT controls, reflecting the complicated relationship between inflammation and OA according to species [122]. Molecular features of cellular aging include genomic instability, dysregulated nutrient sensing, loss of proteostasis, and mitochondrial dysfunction [123]. Senescent cells (SnCs) exhibit proinflammatory secretome, known as the senescence-associated secretory phenotype (SASP) that indices degenerative changes in cells and tissues [124]. Chondrocytes respond to oxidative stress by upregulating p53 and p21 expression and activating p38 mitogen-activated protein kinases (MAPK) and phosphorylation of phosphatidylinositol 3-kinase (PI3K)/Akt signaling pathways, which in turn stimulates a SASP [125,126]. ACLT in mice led to the accumulation of SnCs in the articular cartilage and synovium, and selective elimination of these cells attenuated the development of post-traumatic OA [127]. Remarkably, pain was relieved in the joints soon after SnC clearance, before any tissue structural changes occurred, which implies that cellular senescence drives pain of OA, independent of joint damage.

Although female sex is a risk factor for both OA and OA-associated pain, female mice have been infrequently used for the study of OA [128], with only 20% to 30% of studies using female mice and many studies using only male mice. The reasons for this sex imbalance in OA animal models are speculative. A study that used DMM mice showed that male mice develop more severe OA than female mice, suggesting that sex is a key factor in the progression of OA [129]. In addition, OA models using STR/ort and IL-6 KO mice have shown a higher prevalence of OA in male than female mice [122,130]. A study that used the CIOA mouse model showed sex-related differences in the prevalence of cartilage damage in addition to differences between mouse strains [131]. On the other hand, we have shown that joint damage and pain behavior develop similarly after DMM in both male and female C57/BL6 mice [132]. However, TRPV-1 antagonist capsazepine significantly reduced DMM-induced pain and the expression of TRPV-1 in DRG only in male mice [103]. In another study that used DMM in C57BL/6 mice, although OA changes were evident in 12-month-old females, the extents of cartilage degradation, subchondral bone plate sclerosis, and osteophytes were milder than in males [78]. The development of OA changes after DMM surgery depends on the activity level of the animals, which may explain this discrepancy. As pain is such a prominent feature that shows sex-related differences, not only in OA, but also in other illnesses, it is important to study the mechanisms underlying the sex differences in pain [133]. Proper use and representation of female mice in OA pain research is urgently needed to be able to elucidate sex differences in this debilitating disease (Figure 1).

## 3. Rheumatoid Arthritis

### 3.1. Pain in Clinical RA

Compared to OA, which is considered a type of noninflammatory arthritis, research endeavors in RA have focused on the regulation of abnormal immune responses and inflammation, with pain considered a byproduct of inflammation that would be controlled by reducing disease activity. Although biological disease-modifying anti-rheumatic drugs (DMARDs) have revolutionized disease control and long-term outcomes of RA, substantial numbers of patients still suffer from pain despite low disease activity. The use of biological DMARDs has provided new insights into the unique qualities of pain manifestations independent of inflammation in RA [134]. The blockade of TNF-α improves pain faster than the resolution of inflammation or tissue damage, indicating a direct role for TNF-α in nociceptor sensitization pathways [135]. This direct analgesic effect is not shared by all cytokine blockers, as neutralization of IL-1β has a less pronounced influence on pain despite its profound role in attenuating inflammation and structural damage [136]. In a study of nociceptive central nervous system (CNS) activity using functional MRI in RA patients treated with a monoclonal antibody to TNF-α, neuronal activity in the thalamus and the somatosensory cortex, areas typically involved in pain perception, was significantly reduced as early as 24 h after the infusion [137]. This indicates that pain in RA is mediated by a mechanism independent of inflammation or joint damage. Pain is an important contributor to the patient global assessment of RA disease activity and discordance between patients and physicians [138].

There has been some speculation on the mechanism underlying persistent pain despite the absence of inflammation. It may result from irreversible joint damage and from changes in the CNS processing signals from the joint [139]. Central sensitization is present in people with RA, such that the prevalence of patients fulfilling fibromyalgia classification increases throughout the course of the disease [140].

Conditioned pain modulation, a measure of inhibitory pathways leading to a diffuse decrease in pain in response to acutely painful stimuli, is impaired in RA patients, and this association is mediated by sleep problems [141]. Patients with established RA also show increased sensitivity to pressure-induced pain at non-joint sites, such as the sternum and anterior tibia, as well as over joints, further indicating that central mechanisms contribute to pain in RA [142]. In a study of 108 RA patients, electrophysiological evidence of neuropathy, including pure sensory or sensory motor axonal neuropathy and demyelinating neuropathy, was present in 57.4% of cases [139]. Local inflammatory responses arising from synovitis as well as systemic inflammation caused by circulating proinflammatory cytokines may augment central pain processing in RA. The potentiation of anti-rheumatic medication is indicated mostly to control inflammation, and a high disease activity score driven by pain should be distinguished to avoid over-treatment. A better understanding of the mechanism of such pain is thus of paramount importance.

### 3.2. Animal Models for Studying the Pathogenesis of RA Pain

#### 3.2.1. Animal Models

Animal models of RA are divided into immunization and transfer models [143]. The former is induced by active immunization with normal joint constituents, such as type II collagen (CII) or proteoglycan, and encompass both innate and adaptive immune responses. The collagen-induced arthritis (CIA) model, the most commonly used animal model of RA, is induced by intradermal injection of heterologous CII in complete Freund’s adjuvant (CFA) into the tail [144], leading to an autoimmune response in the joint, such as increased infiltration of circulating immune cells, chronic inflammation, and tissue damage, mimicking human RA [145]. Inflammatory polyarthritis is induced in specific mouse strains, such as DBA1 and C57BL/6. In antigen-induced arthritis (AIA), injection of methylated bovine serum albumin (mBSA) mixed with CFA into the base of the tail or knee joint of mice or rats leads to monoarticular arthritis. The subsequent pathology includes immune complex-mediated inflammation followed by articular T cell-mediated responses [146]. On the other hand, the transfer model is induced by injection of pathogenic autoantibodies or serum, and recapitulates the effector phase of RA. The collagen antibody-induced arthritis (CAIA) model is induced by injecting a cocktail of anti-CII antibodies and lipopolysaccharide (LPS), and is applicable independent of mouse strain or genotype [147]. The K/BxN serum-transfer arthritis model is induced by injecting anti-glucose-6-phosphate isomerase (anti-GPI)-positive serum from K/BxN mice into commonly used mouse strains, leading to pathological changes resembling human RA. Both models circumvent the induction phase of RA, and lead to joint inflammation and destruction of articular cartilage as well as elevated inflammatory markers in serum. In addition, a variety of genetically modified knockout or transgenic mice, such as IL-1 receptor antagonist knockout (IL-1RA-KO), IL-6 receptor (IL-6R) knock-in, and TNF-α transgenic (Tg) mice, were used to study the influence of molecular mediators in inflammation and to test the efficacy of novel therapeutics [143,144,148,149,150,151].

#### 3.2.2. Behavioral Tests to Assess Pain in RA Animal Models

Experimental methods for assessing pain behaviors in RA models are largely similar to those used for OA models, although there are discrepancies in response patterns. In addition, because RA models generally exhibit polyarthritis, modalities useful for measuring asymmetric joint pathology sometimes cannot be used for RA models. Compared to OA, RA models exhibit a greater degree of joint inflammation, which is reflected in their pain behaviors. Mechanical allodynia measured with von Frey fiber and thermal hypersensitivity measured with hot plate response time tend to appear earlier in RA models compared to OA models. While static weight bearing is useful for pain assessment in OA models, because of its polyarticular nature, it is not appropriate for assessment of many RA models. It is notable that in human TNF-Tg (hTNF-Tg) mice, gait parameters measured using catwalk analyses have revealed the association between gait abnormalities and the extent of cartilage damage and bone erosions, but not with the extent of synovitis [152]. These observations indicate that functional impairment in animal models of RA is dependent more on joint destruction than on inflammation.

Both the CIA and CAIA models clearly exhibit mechanical allodynia and thermal hypersensitivity in the early phase of arthritis [153]. While mechanical hypersensitivity persists until the late phase, thermal hyperalgesia tends to decrease to the basal level in the late phase [153]. Moreover, the CIA and CAIA models also display changes in spontaneous behavior, such as reduced climbing, locomotion, and grooming activities [153]. The AIA model displays pathological changes in a single joint, as in OA models, while pain behaviors measured using von Frey or incapacitance tests occur earlier compared to OA models [154]. In this model, mechanical allodynia, thermal hypersensitivity, and abnormal incapacitance tests are normalized in the later phase [154]. The K/BxN model exhibits persistent pain with mechanical hypersensitivity, which does not return to baseline levels during disease progression and outlasts inflammation [155]. Table 2 briefly summarizes the pathology and pain behaviors reported in RA models [134,137,153,154,155,156,157,158,159,160,161,162,163,164,165,166,167].

#### 3.2.3. Understanding the Molecular Mechanism Underlying the Pathogenesis of RA Pain via Experiments in Animal Models

Compared to research on the pathogenesis of aberrant immunity and inflammation, the mechanism of pain in RA animal models has been explored only recently (Figure 2).

Although pain behavior largely parallels the development of inflammation, as in human patients, it is not always concordant with the degree of inflammation. As in humans, pain outlasts inflammation in monophasic RA models induced with antibodies. In the K/BxN serum transfer arthritis model, mechanical allodynia develops congruent with joint swelling and persists after the resolution of inflammation [155]. A transition from an inflammatory to a neuropathic pain state is indicated because TNF and prostaglandin inhibitors alleviate allodynia at the peak of joint inflammation, while only gabapentin relieves allodynia following its resolution [126]. Activating transcription factor 3 (ATF-3), a marker of nerve injury, is significantly increased in the lumbar dorsal root ganglia during the late phase, further corroborating the development of neuropathic pain [126]. TLR-4, a receptor mediating innate immunity and response to DAMPs, has been implicated as a mediator of pain, such that TLR-4 KO mice display a significant reversal of mechanical hypersensitivity and diminished appearance of glial activation markers after the resolution of peripheral inflammation induced by K/BxN serum transfer [168]. In the CAIA model, pain exists prior to, and outlasts, the visual signs of inflammation [169]. As in KBxN serum transfer arthritis, a lack of antinociceptive effects of diclofenac in the post-inflammatory phase of arthritis, and time-dependent activation of spinal astrocytes and microglia, are observed. In a study on a mouse model of CIA, significant reductions in both mechanical and thermal pain thresholds were detected on the day of onset of clinical arthritis [170]. While mechanical pain thresholds remained significantly reduced compared to naive mice for up to 28 days after the onset of arthritis, thermal thresholds returned more quickly to the levels observed in naive mice. In a study on a rat model of CIA, significant mechanical hypersensitivity developed before the onset of clinical signs of arthritis, consistent with observations in patients with RA, who often develop pain ahead of overt inflammatory signs [171]. In that study, significant microglial and astrocytic responses, alongside T cell infiltration, were observed in the spinal cord, and intrathecal delivery of a cathepsin S (Cat S) inhibitor and a fractalkine neutralizing antibody, microglial inhibitors, attenuated mechanical hypersensitivity. In another study, the role of proinflammatory cytokines in the generation of pain was investigated in AIA model induced by mBSA. Immunized mice showed mechanical hypernociception, which was abrogated by an antibody against IL-17 [172]. In another study, IL-1R type I (IL-1RI) expression in lumbar DRGs was significantly upregulated in AIA rats, and although treatment with anakinra did not significantly reduce the severity of arthritis or mechanical hyperalgesia, it reduced thermal hyperalgesia [173]. Anakinra treatment downregulated the expression of TRPV-1, which was accompanied by a pronounced reduction in thermal hyperalgesia. Mechanical hyperalgesia in an AIA model was paralleled by the upregulation of phospho-CREB (pCREB) in the lumbar DRG neurons [174]. This was prevented by IL-1 inhibitor, but not by TNF neutralization, suggesting again the importance of IL-1β in neuroplastic changes in sensory neurons in inflammatory arthritis. In a study on a mono-arthritic multi flare rat streptococcal cell wall (SCW) model, in which a local injection of SCW caused the rapid onset of inflammation and pain, prophylactic administration of etanercept inhibited paw swelling and pain [175]. However, the extent of inhibition of pain was less than that of inflammation. The mechanism underlying the dissociation between inflammation and pain has been explored in Fc gamma receptor 1 (FcγR1) KO mice [176]. In a study that employed AIA and CFA models, FcγR1 signaling was upregulated in joint sensory neurons, while the blockade or global genetic deletion of FcγR1 significantly attenuated arthritis pain and hyperactivity of joint sensory neurons without changes in joint inflammation [176]. Antibodies specific for CII or cartilage oligomeric matrix protein (COMP) were found to elicit mechanical hypersensitivity in mice independent of inflammation [160]. Interestingly, CII-immune complex and CII antibodies did not induce mechanical hypersensitivity or pain-like behavior in FcγR chain-deficient mice, suggesting functional coupling between autoantibodies and pain transmission. In murine glucose-6-phosphate isomerase (G6PI)–induced arthritis, nonremitting arthritis induced by depletion of regulatory T cells led to more severe bone destruction and more persistent thermal hyperalgesia [177]. Up-regulation of the neuronal injury marker ATF-3 in sensory neurons ahead of clinical inflammation indicated an early and persisting affection of sensory neurons by G6PI-induced arthritis [177]. These results indicate that the mechanism underlying pain in RA animal models precedes inflammation.

Inflammation and suppression of pain-related spontaneous behaviors, such as locomotor activity, operant responses, and place avoidance, are strongly correlated in rapid-onset models of arthritis [134]. However, the more slowly progressing K/BxN mouse strain shows a significant delay between peak clinical progression and decreased mobility [178]. This finding is suggestive of a psychoaffective component to pain-suppressed behaviors, which is commonly observed in human RA.

## 4. Conclusions

Recent advances in animal experimentation and molecular biology have led to significant progress in our understanding of the mechanisms underlying the pathogenesis of arthritis pain. The concept that pain is a mere byproduct of joint damage and nociceptive response has become outdated. Aside from DAMPs produced from joint destruction, various nociceptive players, such as NGF- TrkA axis, CGRP, TRPV-1, and SASP, have been proposed as molecular mediators of OA pain. For RA, pain independent of inflammation has been correlated with ATF-3, TLR-4, and FcγR1. With the advances in cutting edge molecular techniques such as single cell polymerase chain reaction (PCR) and CRISPR/Cas9 mediated gene regulation, the knowledge base for pain pathogenesis would be significantly and rapidly expanded. Despite the inherent difficulty in extrapolating data gained in animal pain behavior studies to human arthritis, the critical assessment of various molecular mediators and translational studies would help to define the biological relevance of novel therapeutic targets for the treatment of arthritis pain.

## Figures and Tables

**Figure 1 ijms-21-00533-f001:**
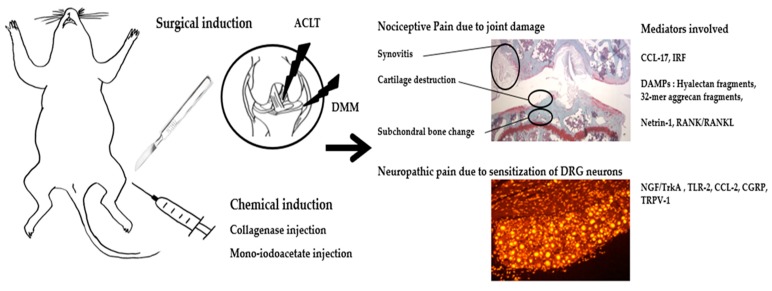
Summary of molecular mediators of pain in OA animal model. ACLT, anterior cruciate ligament transection; DMM, destabilization of the ligament transection; DRG, dorsal root ganglion; CCL-17, chemokine (c-c motif) ligand 17; IRF, interferon regulatory factor; DAMPs, damage associated molecular patterns; 32-mer, 32-animo-acid; RANK/RANKL, receptor activator of nuclear factor kappa-B/RANK ligand; NGF/TrkA, nerve growth factor/tropomyosin receptor kinase; TLR-2, toll-like receptor 2; CCL-2, chemokine (c-c motif) ligand 2; CGRP, calcitonin gene related peptide; TRPV-1, transient receptor potential cation channel subfamily V member 1.

**Figure 2 ijms-21-00533-f002:**
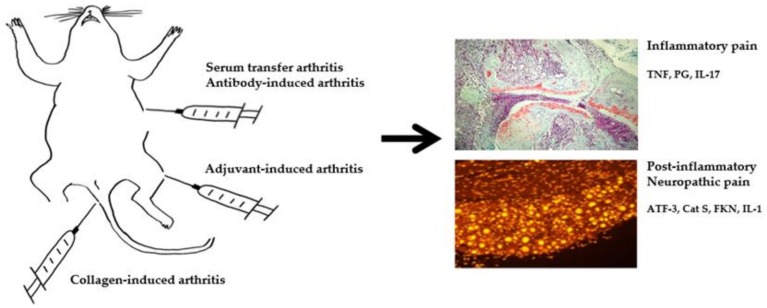
Summary of molecular mediators of pain RA animal model. TNF, tumor necrosis factor; PG, prostaglandin; IL-17, interleukin 17; ATF-3, activating transcription factor 3; Cat S, cathepsin S; FKN, fractalkine; IL-1, interleukin 1.

**Table 1 ijms-21-00533-t001:** Common OA models and reported pain behaviors.

Type	Model (Onset of Pathology)	Pros/Cons	Pathological Findings	Behavioral Assay (Pain Response Onset)	Findings	Molecular Pathogenesis Identified
Surgical	DMM (4 wk)	Mimics human post-traumatic OA, slow disease progression and mild cartilage damage, useful in assessing therapies. Need specially trained surgeon, risk of infection.	-Cartilage degradation -Synovial hyperplasia -Bone sclerosis -Osteophyte formation	von Frey (4 wk), incapacitance (4 wk), hot plate (8 wk), LABORAS (8 wk), gait abnormality (10 wk)	Pain response is progressively induced in early phase. Most representative method: von Frey	Loss of PKCδ exacerbates pain in DMM. ADAMTS-5 inhibition attenuates pain in DMM. Knockout of RANKL inhibits pain in ACLT mice. Knockout of Netrin-1 inhibits pain in ACLT mice.
	MNX (2 wk)	More rapid disease onset and higher damage in the joint compared with human disease, useful in assessing therapies. Need specially trained surgeon, risk of infection.		von Frey (1 wk), incapacitance (3 d), cold plate (5 wk)		
	ACLT (2 wk)			von Frey (1 wk), incapacitance (1 wk), hot plate (4 wk), rotarod (4 wk), LABORAS (1 wk), gait (8 w)		
Chemical	MIA (3–7 d)	Easy local injection method, rapid induction of severe joint degeneration, useful for studies in pain behavior. More difficult to translate to human setting.	-Inhibition of glycolysis and disruption of chondrocyte metabolism -Cartilage degradation -Synovial hyperplasia -Osteophyte formation	von Frey (1 wk), incapacitance (3 d), hot plate (6 d), rotarod (15 d), gait (6 d), LABORAS (14 d)	Pain response is induced rapidly within a week post-injection. Most representative methods: von Frey and incapacitance	Neutralization of CCL-17 ameliorates pain in CIOA. Increase of IRF-4, CCL-17 in CIOA.
	CIOA (1–4 wk)	Most rapid progression, useful in assessing therapies. More difficult to translate to human setting.	-Cartilage degradation -Osteophyte formation	von Frey (1 wk), incapacitance (1 wk), hot plate (1 wk)		
Spontaneous/noninvasive	STR/ort (8–16 wk)	Might mimic primary human OA, no specially trained personnel required. Long period for the development of OA, high variability of the disease phenotype and incidence.	-Cartilage degradation -Osteophyte formation	von Frey (no difference), gait (20 w), cold plate (no difference)	Pain response does not show significant variation with age.	STR/ort mice do not show any signs of pain even when treated with the opioid antagonist naloxone.
	Mechanical joint loading (1 wk)	Noninvasive, suitable to study the effects of mechanical loading and intra-articular fracture, rapid induction of severe joint degeneration. Need specialized equipment.	-Cartilage degradation -Bone sclerosis -Osteophyte formation	von Frey (2 wk), incapacitance (4 wk), hot plate (no difference), rotarod (5 wk)	Pain response is progressively induced in early phase.	Anti-NGF alleviates pain in mechanical joint loading model.

DMM, destabilization of medial meniscus; MNX, medial meniscal transection; ACLT, anterior cruciate ligament transection; MIA, monosodium iodoacetate; CIOA, collagenase-induced arthritis; OA, osteoarthritis; LABORAS, laboratory animal behavior observation registration and analysis system; d, day; wk, week(s); PKCδ, protein kinase C delta; ADAMTS-5, a disintegrin and metalloproteinase with thrombospondin motifs 5; RANKL, receptor activator of nuclear factor kappa-B ligand; CCL-17, chemokine (c-c motif) ligand 17; IRF-4, of interferon regulatory factor 4; NGF, nerve growth factor.

**Table 2 ijms-21-00533-t002:** Common RA models and reported behaviors.

Type	Model (Trigger, Onset of Pathology)	Pros/Cons	Pathological Findings	Behavioral Assay (Pain Response Onset)	Findings	Molecular Pathogenesis Identified
Induced (onset after first immunization)	CIA (CII/adjuvant)	Most common induced model, RA-like pathogenesis. Restricted strains in mice, severe progressive disease.	-Adaptive immune system activation against endogenous joint epitopes. -Inflammation -Immune cell infiltration -Joint destruction and synovial hyperplasia	von Frey (30 d), rotarod (5 d), locomotion (25 d), tail flick (1 wk), Hargreaves (24 d)	Pain response is induced rapidly after induction. Pain response increases in early phase and persists. Most representative methods: von Frey and Hargreaves	Anti-Cat S and FKN attenuate pain in CIA. Gabapentin and buprenorphine attenuate pain in CAIA. Knockout of TLR-4 inhibits pain in K/BxN.
	CAIA (Anti-CII Ab)	Efficient and robust to study the effector phase of RA, diverse susceptible strains. Does not involve full spectrum of immune activation.	-Inflammation -Immune cell infiltration -Joint destruction	von Frey (6 d), hot plate (15 d), locomotion (5 d)		
	K/BxN (Serum/Anti-GPI Ab)			von Frey (2 d), Hargreaves (3 d), locomotion (3 d)		
	AIA (mBSA/adjuvant)	Local RA-like pathogenesis, localized inflammation. Damage to cartilage less severe than in RA.	-Adaptive immune system activation against exogenous epitopes -Inflammation -Immune cell infiltration -Joint destruction and synovial hyperplasia	von Frey (22 d), Hargreaves (7 d), gait (15 d), incapacitance (17 d)	Pain response is induced rapidly after induction. Pain response increases in early phase and restores in late phase. Most representative methods: von Frey and incapacitance	Anti-IL-17 alleviates pain in AIA. Increase of IL-1RI and pCREB in AIA.
Spontaneous	TNF-Tg (hTNF overexpression)	Useful for studies in effect of excess TNF in RA. Only been identified in mice, does not involve full spectrum of immune activation.	-Inflammation -Immune cell infiltration -Joint destruction and synovial hyperplasia -Pannus formation	von Frey (6 wk), tail flick (10 wk), Hargreaves (6 wk)	Pain response increases in early phase and persists. Most representative methods: von Frey and Hargreaves	Increase of the nociceptive brain activity in TNF-Tg mice.
	IL-1RA−/− (Genetic deficiency of 1L-1Ra)	Useful for studies in effect of IL-1 signaling in RA. Only been identified in mice.		No behavioral data available to date.		

CIA, collagen-induced arthritis; CAIA, collagen antibody-induced arthritis; AIA, antigen-induced arthritis; TNF-Tg, tumor necrosis factor transgene; IL-1RA–/–, interleukin-1 receptor antagonist knockout; CII, collagen type II; Ab, antibody; mBSA, methylated bovine serum albumin; GPI, glucose-6-phosphateisomerase; hTNF, human tumor necrosis factor; RA, rheumatoid arthritis; d, day; wk, week(s); Cat S, cathepsin S; FKN, fractalkine; TLR-4, toll-like receptor 4; IL-1RI, interleukin-1 receptor type I; pCREB, phospho-CREB.

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
