# Peer review of "Understanding the Molecular Mechanisms Underlying the Pathogenesis of Arthritis Pain Using Animal Models"

_ijms, 2020, doi:10.3390/ijms21020533_

Round 1
Reviewer 1 Report
The topic in question is relevant. The review is rather succinct, as it has been analyzed from 146 sources.
I wonder what keywords were used to find literature on the topic in question. Please, describe.
I would like to see a broader overview of the relationship between pain and age and sex in both RA and OA. Please, describe.
Reviewer 2 Report
This is a good paper about pathology of arthritis pain using animal models .
However this review is not usefullness, everybody knows things that authors describes this paper except beginners.
Reviewer 3 Report
This is an well written, interesting and timely review of the use of animal models to investigate the molecular mechanisms underlying the pathogenesis of arthritis pain, which covers osteoarthritis (OA) and rheumatoid arthritis (RA). However, at present I feel that the manuscript is a little unbalanced. As the title of the manuscript mentions specifically molecular mechanisms I think that further, in-depth analysis of molecular processes in the OA portions of the manuscript is needed, particularly in the section ‘Understanding the molecular mechanism underlying the pathogenesis of OA pain via experiments in animal models’. Much of the discussion that is included focuses on aetiological factors (age, obesity, sex) or whole cell involvement (osteoclasts) rather than molecular level analysis.
Furthermore, in some cases (highlighted below) the body of work selected for discussion could perhaps be a little more updated. This is particularly true since, as the authors point out, recent advances in animal experimentation and molecular biology have provided significant insights into this area. This is of particular relevance in the area of pain measurements, which are historically difficult to make accurately in animal models. The discussion itself might be more informative to the reader if suggestions were made on which model selection/ pain measurement modalities are advantageous for the molecular analysis performed or targeted to be performed based on the reviewed literature. To aid the reader I feel a summary figure or diagram of how determined molecular networks interact to drive pain in OA and RA might be beneficial, perhaps as related to disease model/ stage.
Specific Comments
Title- I would suggest making this ‘mechanisms’ rather than mechanism.
Introduction
Line 31-32 “Furthermore, YLD due to OA increased by 64% from 1990 to 210…”- Can more up-to-date information be included here as there is approximately a decade of missing information that could be informative for the reader. Perhaps the contribution of RA to worldwide YLD could be also included.
Line 39-40 “The lack of therapeutics for optimal pain management is partially responsible for the current epidemics of opioid and narcotic abuse” and the following sentences– Is this for all types of pain or OA/RA? Inclusion of perhaps a reference to this effect stating that opioid prescription for OA/RA is a major driver of this epidemic. This is a complex socioeconomic issue, it might be interesting to the reader if a global statistic or perhaps a couple of regional ones outside of the USA were very briefly mentioned and cited here as well to show the universal link.
Line 46-48 ‘However, in addition to nociception, arthritis pain involves diverse mechanisms,
including processing of pain in the nervous system as well as psychological distress”. – Please add references for this.
Overall- The introduction jumps between OA and RA studies frequently for supporting statistics. While it is important to discuss both, perhaps dealing with one then the other for each point would be clearer for the reader.
OA
Pain in clinical OA
Final paragraph- Some of these studies were conducted some time ago- have any more recent studies examined this, particularly as the author state there is debate over the processes involved?
Animal models
This section provides an overview of the types of animal model of OA, perhaps the pros/cons of these surgical, chemical and mechanical approaches to OA induction could be summarised in a table with respect to usefulness for probing pain in OA at particular molecular/ system levels. The authors state that ‘selection of a suitable animal model is required to investigate progression of the disease and pain’- by summarising previously reported pros and cons this section could aid the reader in this regard.
Line 135- ‘a suitable animal model is required’ is a different font size in my copy.
Behavioural tests to assess pain in OA animal models
Line 155- ‘10 min’ should say ‘10 minutes’.
Are these behavioural tests equally effective in determining animal pain
Understanding the molecular mechanism underlying the pathogenesis of OA pain via experiments in animal models
As this section directly relates to the title of the review I suggest that a deeper discussion of molecular events, markers, signalling pathways and targets/effectors is needed in addition to nice descriptions provided.
Paragraph 1- This section provides a nice summary of comparative papers of pain induction between methods of OA induction, particularly the role of bone resorption in the latter parts of the first paragraph in the section. However, aside from the CRISPR reference [83] there is little discussion of the ‘molecular mechanisms’ underlying pain or the differences between models. What do the differences in pain onset tell us? Again several of these studies are over 10 years old, it would be interesting for the reader if new molecular biology/genetic/ bioimaging techniques have generated new information in this area building on these earlier studies.
Obesity paragraph- While an interesting summation of the role of obesity, a molecular level discussion is only briefly entered into with mention of leptin signalling and IL-12p70 production. As in the previous section, references are quite old- do more recent studies investigate on a molecular level the findings reported here?
Line 181 ‘Understanding the molecular mechanism underlying the pathogenesis of OA pain via experiments in animal models’ – This title should be italicised.
Age paragraph, line 235- As per the previous two sections, a discussion of molecular level changes related to aging is missing.
Sex paragraph, line 247- as the previous sections.
This section might benefit from a discussion of how animal models can/ could be used to determine molecular roles in OA pain. CRISPR mice would be one example of this but other tools exist and promising techniques could be mentioned here, ie what is happening at the cutting edge.
Furthermore, a small summary of the findings of the above discussion could be added as a final paragraph perhaps highlighting key/ likely key molecular players, promising drugable targets, the most useful/ promising animal models, approaches currently being used developed as determined by the authors from their review of the current literature.
RA
Line 274- ‘TNF’ should read TNF-a
Line 275-IL-1 should be (I presume) IL-1b
Line 355 – ‘Understanding the molecular mechanism….’ This subheading should be in italics.
Line 363 – ‘A transition from an inflammatory to a neuropathic pain state is indicated because TNF and prostaglandin inhibitors alleviate allodynia at the peak of joint inflammation, while only gabapentin relieves allodynia following its resolution’- is this finding and the subsequent sentence from the reference above this statement [126] or below? Could you add in a citation to clarify this.
Line 404- ‘These results elucidate the mechanism underlying the functional coupling between autoimmunity and pain generation independent of inflammation’- I am not sure the results presented here do completely elucidate this. Perhaps a short summary of the results described and how they fit together in a broader sense could be added at the end of this section prior to the conclusion
Conclusion
A more specific concluding statement could be added for OA and for RA on where we currently are in using animal models for molecular understanding of pain.
